# Fast and Accurate Approximation Methods for Trigonometric and Arctangent Calculations for Low-Performance Computers

**Takashi Kusaka** [1,*,†] and **Takayuki Tanaka** [2,†]

1  Independent Researcher, Sapporo 063-0867, Japan
2  Graduate School of Information Science and Technology, Hokkaido University, Sapporo 060-0814, Japan; ttanaka@ssi.ist.hokudai.ac.jp
*  Correspondence: kusaka@frontier.hokudai.ac.jp
†  These authors contributed equally to this work.

**Abstract:** In modern computers, complicated signal processing is highly optimized with the use of compilers and high-speed processing using floating-point units (FPUs); therefore, programmers have little opportunity to care about each process. However, a highly accurate approximation can be processed in a small number of computation cycles, which may be useful when embedded in a field-programmable gate array (FPGA) or micro controller unit (MCU), or when performing many large-scale operations on a graphics processing unit (GPU). It is necessary to devise algorithms to obtain the desired calculated values without an accelerator or compiler assistance. The residual correction method (RCM) developed here can produce simple and accurate approximations of certain nonlinear functions with minimal multiply–add operations. In this study, we designed an algorithm for the approximate computation of trigonometric and inverse trigonometric functions, which are nonlinear elementary functions, to achieve their fast and accurate computation. A fast first approximation and a more accurate second approximation of each function were created using RCM with a less than 0.001 error using multiply–add operations only. This achievement is particularly useful for MCUs, which have a low power consumption but limited computational power, and the proposed approximations are candidate algorithms that can be used to stabilize the attitude control of robots and drones, which require real-time processing.

**Keywords:** arctangent approximation; trigonometric functions approximation; elemental function approximation; fused multiply–add; algorithm design and analysis; computational cost reduction

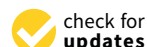



## 1. Introduction

The approximation of elementary functions is a critical issue in science and engineering, including computer graphics, scientific computing, and signal processing [1–4]. Central processing units (CPUs) have evolved to have computation instructions for efficient processing, but electronic circuits with limited resources, such as field programmable gate arrays (FPGAs) and micro controller units (MCUs), may not have accelerators such as a floating-point unit (FPU) or digital signal processor (DSP). Therefore, techniques to avoid calculation with a large number of division and non-linear calculation cycles are important for large-scale calculations and embedded systems [5]. Division and square root are frequently used, even though they are up to ten times slower than addition and multiplication, and this trend has not changed in the latest architectures [6,7].

In particular, MCUs have a low computational power with an instruction set limited to basic arithmetic and logic operations, but they are small in size, have a low power consumption, and are used in a wide variety of products, including smartphones, audio accessories, video game peripherals, and advanced medical equipment [8–13]. Furthermore, a small size and low power consumption features are required in areas such as IoT devices [14,15] and drone control [16–19].

Even in the rapidly developing field of machine learning, fast approximate computation is critical to solve problems such as short learning times and the real-time processing of inferences [20–23]. GPUs can process a large number of multiply–add operations in parallel, and by replacing nonlinear calculations used in machine learning with approximate calculations using multiply–add operations, it may be possible to achieve higher speeds and smaller models.

When the compiler optimization performance was low, implementation techniques to avoid division were very useful in the field of embedded electronic circuits [24]. Even today, many MCUs do not have an extended instruction set, so such techniques are very important. For example, replacing division by a product of reciprocal constants and utilizing bit shifting for multiplying and dividing powers of two are well-known techniques [25]. Computational tricks using the floating-point structure of the IEEE754 are effective, and useful techniques are known for obtaining fast approximations of exponential and logarithmic functions [21,26] and inverse square roots (reciprocal sqrt) [25,27–32]. In particular, the latter is a well-known algorithm called the fast inverse square root (FISR), which is a useful technique that can be used to find the reciprocal and square root.

As mentioned, it is important to use such techniques to obtain elementary functions at a high speed and low cost when considering integration into electronic circuits or use in machine learning with GPUs. Among elementary functions, the addition and multiplication in four fundamental arithmetic operations can generally be processed by CPUs at a high speed, and the division can be calculated by using the fast inverse square root described above. Moreover, exponential and logarithmic functions, which are nonlinear functions of elementary functions, can be obtained using IEEE754 calculation tricks. The remaining elementary functions include the computation of trigonometric and inverse trigonometric functions, for which, some methods have been proposed.

In general, approximations of trigonometric functions are often obtained using Taylor expansions, which are slow to converge and cannot be computed over a wide definition domain without the terms exceeding the seventh order. Inverse trigonometric functions are similarly computed using polynomial approximation [33], rational approximation [34–37], look up table (LUT), and coordinate rotation digital computer (CORDIC) algorithms [38–40], but each of them have disadvantages, such as requiring a large amount of memory and iterative calculations.

To solve this problem, we proposed an algorithm to approximate trigonometric functions (sin and cos) and inverse trigonometric functions (atan2) using only the multiply–add operation [41]. In the proposed method, the approximation is achieved by using one addition and two multiplications. The proposed residual correction method (RCM) is an approximation that minimizes the error over the entire domain, rather than a locally precise approximation such as the Taylor expansion. The atan2 function created by the proposed method is faster than the previously used DSP-trick [42], and the approximation is obtained with fewer errors. The DSP-trick is a suitable algorithm for MCUs, but it must always perform division. Thus, it is a tightly coupled algorithm with division. In contrast, our proposed method can be executed with only multiply–add operations if the norm is known. Therefore, it does not necessarily require division. Hence, it is a loosely coupled algorithm with division. Since the norm of the measured values is known for most of the posture measurement sensors, the proposed method, skipping normalization, is suitable for embedded devices. If normalization is required, the algorithm can be integrated with any normalization method, such as fast inverse square root for MCUs or single instruction, multiple data (SIMD) instructions for advanced computing devices.

In this study, we extend our proposed method to compute even more accurate approximations of trigonometric and inverse trigonometric functions at a similar computational cost.

## 2. Concept of Approximation by Residual Correction Method

We have been developing wearable posture and fatigue measurement devices [43,44] and assistive suits [45–47] using small, low-power MCUs. For this purpose, it is necessary to measure human posture in real time from sensor data using only simple arithmetic and logical operations that are possible with MCUs. Therefore, in our previous research, we developed approximate formulas based on multiply–add operations of trigonometric and inverse trigonometric functions to realize posture calculation on MCUs [41].

We proposed the concept of designing approximate expressions utilizing the residual correction method using simple algebraic expressions with only multiply–add operations, without division operations and iterative convergence algorithms, to achieve efficient calculations for embedded systems.

In a previous study, as an example of approximation using the residual correction method, an approximation formula based on the multiply–add operation of trigonometric functions (sin and cos) and inverse trigonometric functions (atan2) was created.

The following procedure was used to design the approximate formula.

(1). Linear approximation with fixed origin and domain endpoints:

$$f(x) \approx ax + b;$$

(2). Evaluate residuals with approximate target function:

$$e_{res} = ax + b - f(x);$$

(3). Design of residual correction function (RCF) :

$$r(x) \approx e_{res};$$

(4). Approximate formula with residual correction:

$$f_{approx}(x) = ax + b - r(x).$$

The concept of the residual correction method is to express the RCF using only multiply–add operations, which must be explored heuristically. Due to the restriction of using only multiply–add operations, the order of the candidate functions for the RCF is less than a third. The RCF was evaluated in terms of both computational cost and approximation error, and it was shown that, for the nonlinear functions mentioned above, the use of parabolic functions provides an efficient approximation formula that expresses each function using only multiply–add operations.

Although the above method can easily obtain a quadratic approximation formula for a nonlinear function, some applications require an even smaller error. In the next section, a method to reduce the approximated error while maintaining the computational cost by extending the approximation formula with residual correction is proposed.

## 3. Second Approximation by the Residual Correction Method for Trigonometric Functions

In a previous study, the approximations made by the residual correction method for trigonometric functions were as follows [41]:

$$\sin(\theta) \approx s(\theta) = \left(\frac{2}{\pi}\right)^2 \theta(\pi - |\theta|) \tag{1}$$

$$\cos(\theta) \approx c(\theta) = s\left(\frac{\pi}{2} - |\theta|\right) \tag{2}$$

These are of the same form as the famous quadratic approximation for a sine function [48] in $[-\pi, \pi]$, but the error is larger around $\theta = \frac{\pi}{4}$, as shown in Figure 1.

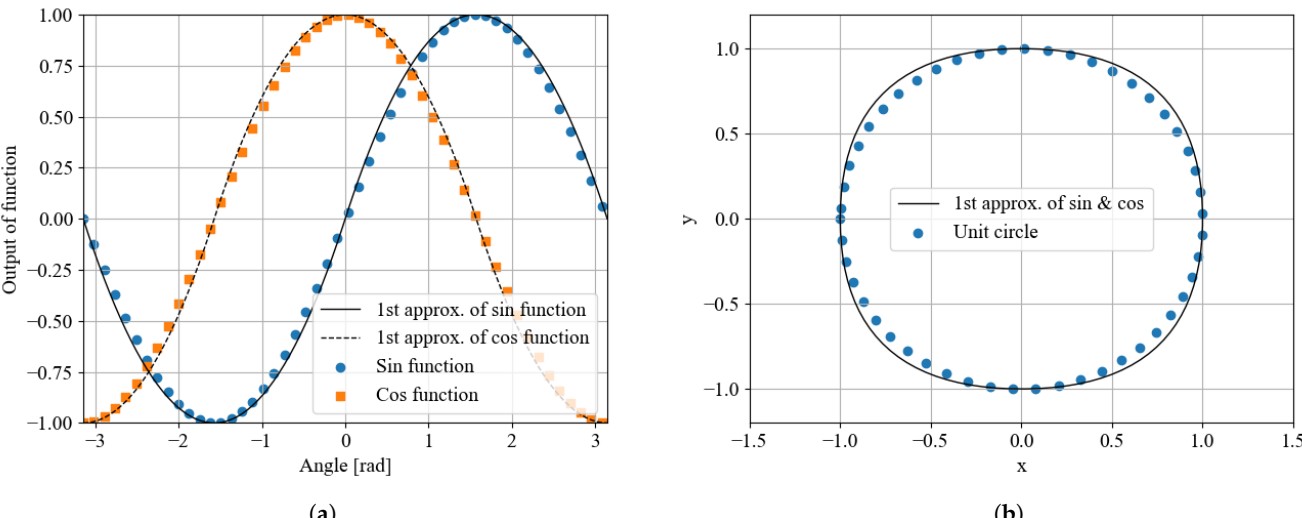

(**a**)            (**b**)

**Figure 1.** Summary of 1st approximations of sin-function and cos-function. (**a**) First approximation of sin and cos functions by RCM. (**b**) Comparison with unit circle.

Here, the above approximation formula is used as the first approximation by the residual correction method, and a method to further reduce the error by utilizing the residual correction method under the constraint of using only multiply–add operation is examined.

### 3.1. Exploring RCF Candidates

Here, we consider the sin-function and evaluate the residuals of the first approximation. The residuals of the first approximation have the shape of an odd function, as shown in Figure 2a. The following RCF candidates obtained by the multiply–add operation can be considered for this residual:

(1). RCF with signed quadratic function:

$$r_{sq}(\theta) = \alpha_{sq} s(\theta)(1 - |s(\theta)|);$$

(2). RCF with cubic function :

$$r_{cubic}(\theta) = \alpha_{cubic} s(\theta)\left(1 - s^2(\theta)\right);$$

(3). RCF with co-function :

$$r_{cofunc}(\theta) = \alpha_{cofunc}\left(s(\theta) - \text{sign}(s(\theta))\sqrt{1 - c^2(\theta)}\right).$$

Although the square root appears in the RCF using the co-function, the built-in implementation will use the fast inverse square root.

Figure 2b shows the evaluation of the RCF candidate functions. Here, the coefficients of the RCF candidate functions are designed as $\alpha_{sq} = 0.224$, $\alpha_{cubic} = 0.145$, and $\alpha_{cofunc} = 0.5$ so that the residuals are well corrected. As can be seen from the graph, the signed quadratic function is closest to the residuals of the first approximation. In addition, the signed process is faster than the multiplication and fast inverse square root, indicating that the signed quadratic function is suitable for RCF.

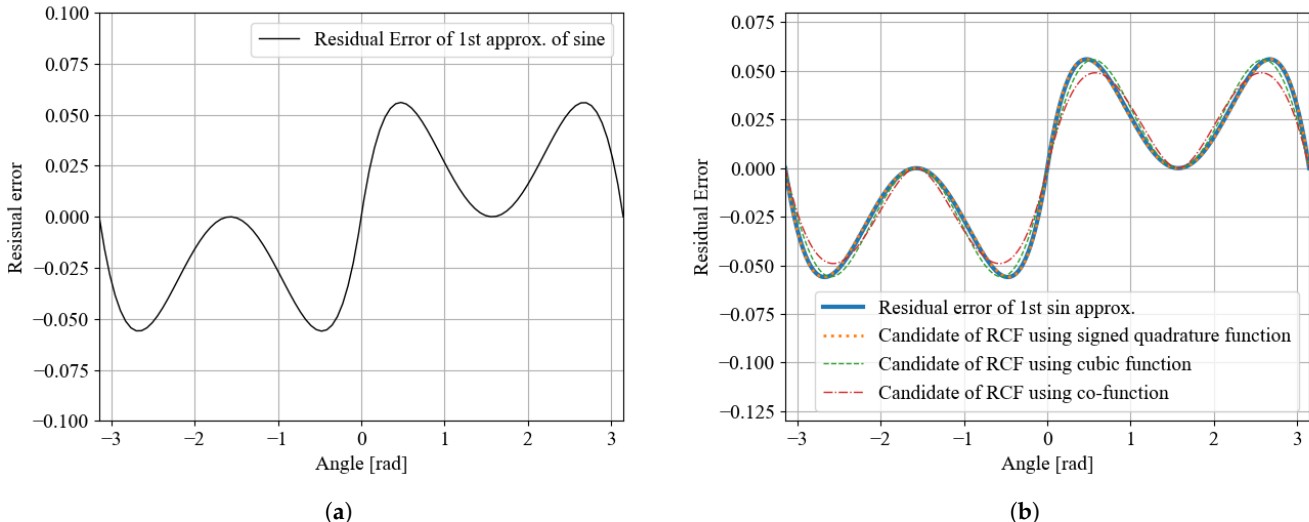

**Figure 2.** Residual error of 1st approximation of sin-function and candidates of its RCF. (**a**) Residual error of 1st approximation of sin function. (**b**) Comparison among designed RCFs.

### 3.2. Second Approximation of Trigonometric Functions

Using $r_{sq}(\theta)$ for RCF, the second approximation of the trigonometric function is

$$
\begin{aligned}
s_2(\theta) &= s(\theta) - r_{sq}(\theta) \\
&= s(\theta)\big[(1 - \alpha_{sq}) - \alpha_{sq}|s(\theta)|\big].
\end{aligned}
\tag{3}
$$

A similar argument can be made for the second approximation of the cos function:

$$
c_2(\theta) = c(\theta)\big[(1 - \alpha_{sq}) - \alpha_{sq}|c(\theta)|\big]
\tag{4}
$$

The approximate shape of the final second approximation function is shown in Figure 3.

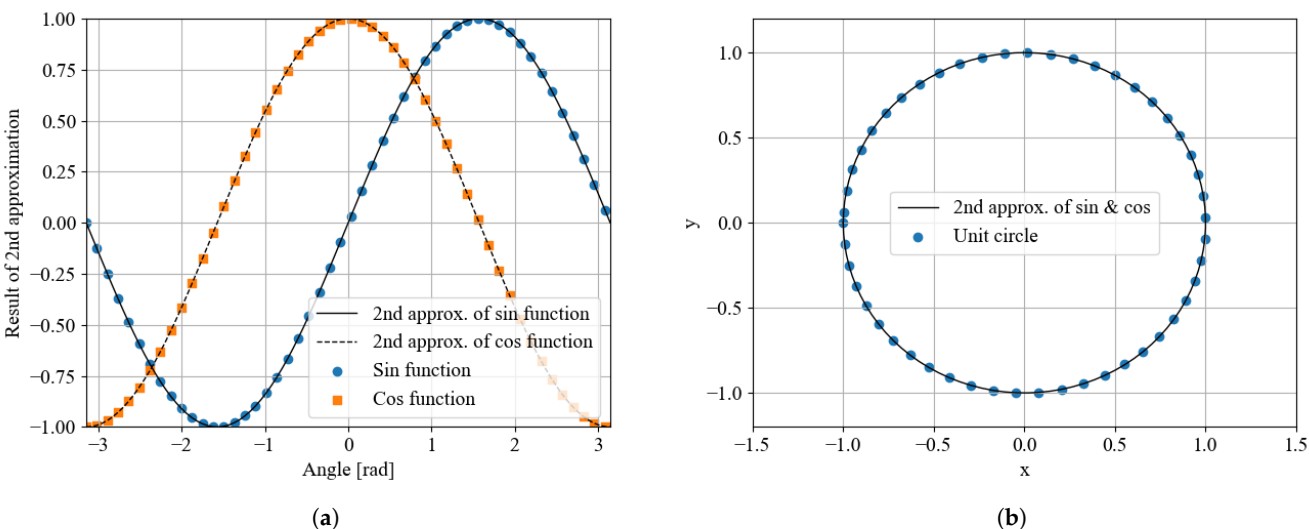

**Figure 3.** Summary of 2nd approximations of sin-function and cos-function. (**a**) Second approximation of sin and cos functions by RCM. (**b**) Comparison with unit circle.

To evaluate the performance of the second approximation, the final residuals and computation time are shown in Figure 4 and Table 1. The final residual is, at most, approximately 0.00092, which is sufficiently small when compared with the first approximation. The computation time was evaluated using an Intel Xeon E3 (2500 MHz) CPU. As a result,

the processing time was 1.4 ns for the sine's approximation and 2.04 ns for the cosine's approximation, which is approximately twice as long as the first approximation, but more than 30 times faster than the built-in functions in math.h.

Comparing the first approximation, Equation (3), with the second approximation, Equation (4), we observe that they have the same parabolic function shape. The second approximation first keeps the results of the calculation of the first approximation in memory, and then uses the results to input them into a parabolic function of the same shape. The first approximation requires the computational cost of one parabolic function, and the second approximation requires the computational cost of two parabolic functions. Therefore, the computation cost is approximately doubled; however, the computation error is reduced to approximately 1/60.

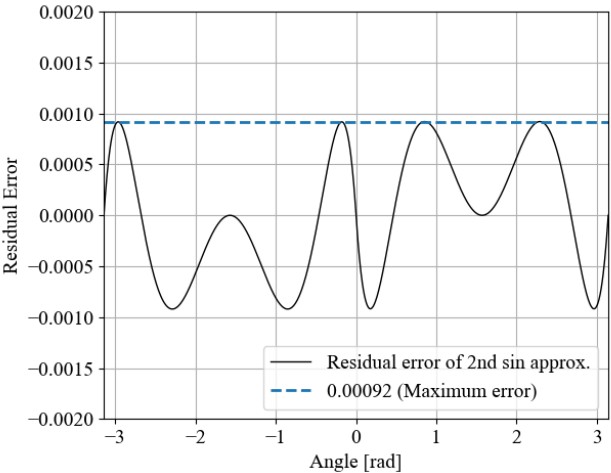

**Figure 4.** Residual error of 2nd approximation of sin-function.

**Table 1.** Calculation time evaluation of 2nd approximations of sin-function and cos-function (Average ± S.D.).

|  | **1st Approx.** | **2nd Approx.** | **Math Library (math.h)** |
|---|---|---|---|
| Sine function | $0.54 \pm 0.16$ ns | $1.4 \pm 0.2$ ns | $68.28 \pm 0.52$ ns |
| Cosine function | $1.22 \pm 0.13$ ns | $2.04 \pm 0.39$ ns | $69.1 \pm 0.90$ ns |

## 4. Second Approximation by the Residual Correction Method for Inverse Trigonometric Functions

Approximate formulas based on the residual correction method for inverse trigonometric functions were also proposed in our previous study [41].

$$\text{atan}(y/x) \approx \text{at}(y, x) = \left( \frac{\pi}{2} - \frac{2}{3}x \right) y \tag{5}$$

Note that this equation is designed assuming that the norm is known, $x^2 + y^2 = 1$, and the domain of definition is $[-\pi/2, \pi/2]$. In many cases, the norm is known for sensors that measure the directional cosine in embedded systems, but when the norm changes occasionally, it is assumed that the norm is normalized in advance using fast inverse square root. The computational complexity of this normalization will be discussed later. Next, we will consider a higher accuracy by the second approximation for inverse trigonometric functions and trigonometric functions.

### 4.1. Domain Expansion

Since the domain of Equation (5) is $[-\pi/2, \pi/2]$, it is necessary to extend the domain in order to treat it the same as atan2$(y, x)$. Two methods are evaluated in this paper: one based on a conditional branch and the other based on an extension using the half-angle identity.

The method using the conditional branch changes the formula used by using the sign of the variable.

$$\text{atan2}(y,x) \approx \text{at2}_{if}(y,x) = \begin{cases} \text{at}(y,x), & \text{if } x \geq 0 \\ \pi - \text{at}(y,-x), & \text{if } x < 0 \text{ and } y \geq 0 \\ -\pi - \text{at}(y,-x), & \text{otherwise.} \end{cases} \tag{6}$$

Another way to expand the domain of definition by a factor of two is to use the half-angle identity. This can be expressed in a single algebraic expression, eliminating the need to switch equations. The atan's half-angle identity,

$$\text{atan2}(y,x) = 2\text{atan}\left( \frac{y}{\sqrt{x^2 + y^2} + x} \right), \tag{7}$$

together with the constraints of $x^2 + y^2 = 1$, can be organized into

$$\text{atan2}(y,x) \approx \text{at2}_{ex}(y,x) = \left( \frac{\pi}{\sqrt{2}} \frac{1}{\sqrt{1+x}} + \frac{2}{3} \right)y \tag{8}$$

$$= \left( \frac{\pi}{\sqrt{2}} G(1+x) + \frac{2}{3} \right)y. \tag{9}$$

Here, $1/\sqrt{\cdot}$ appears in the equation, and $G(\cdot)$ is used as the fast inverse square root. This allows us to extend the definition domain from $[-\pi/2, \pi/2]$ to $[-\pi, \pi]$, so that we can approximate atan2 with a single algebraic expression.

The results of each domain extension are shown in Figure 5a, and the residuals are shown in Figure 5b.

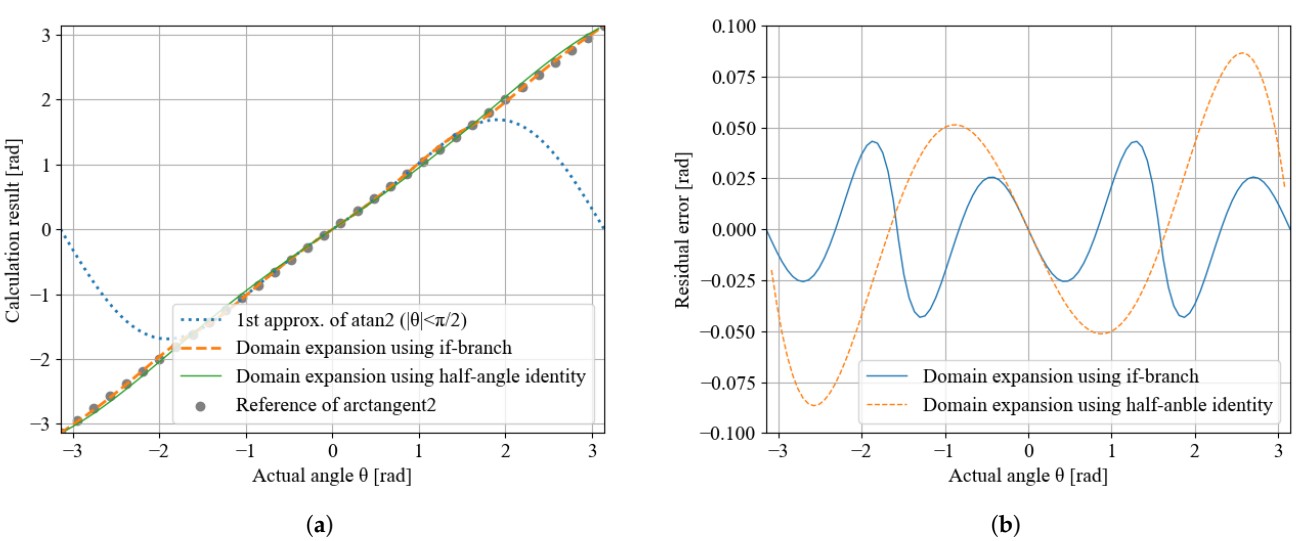

(**a**) 　　　　　　　　　　　　　　　　　　　　　　　(**b**)

**Figure 5.** First approximation of atan2 and its domain expansion (plot using $x = \cos\theta, y = \sin\theta$). (**a**) First approximation of atan2 and domain expansion. (**b**) Residual error in each expansion.

### 4.2. Exploring RCF Candidates

Next, we consider candidates for RCF, but, as the atan2 function is a two-variable function, it is difficult to search for functions by simple multiply–add operations, such as parabolic functions. Therefore, we consider using Newton's method by referring to the steps for improving the accuracy of the fast inverse square root.

From the constraints on the trigonometric functions, the evaluation function $f(\theta)$ and its derivative in Newton's method by $x$, $y$, and $\theta$ are

$$
\begin{aligned}
f(\theta) &= \frac{1}{2}\left[(x - \cos\theta)^2 + (y - \sin\theta)^2\right] \\
&= 1 - x\cos\theta - y\sin\theta,
\end{aligned}
\tag{10}
$$
$$
f'(\theta) = x\sin\theta - y\cos\theta.
\tag{11}
$$

Using this, the error converges to zero by repeating this equation

$$
\theta_{n+1} = \theta_n - \frac{f(\theta_n)}{f'(\theta_n)}.
\tag{12}
$$

With reference to this, the second term on the right-hand side, which is the correction term, can be used as a candidate for RCF.

Thus, it was found that the sin and cos functions are necessary for RCF to cancel the approximation error of atan2. From the evaluation in the previous chapter, the trigonometric functions in the math library are computationally expensive when implemented in programs. Therefore, we replace them with $s_2(\theta)$ and $c_2(\theta)$, which are highly accurate and fast approximate calculations.

Therefore, let the candidate functions of RCF be as follows:

$$
r_{atan2}(y, x) = \frac{1 - xc_2(\theta) - ys_2(\theta)}{xs_2(\theta) - yc_2(\theta)}
\tag{13}
$$

There are two problems with this RCF candidate. One is that it violates the design concept of the residual correction method because it involves division. The other is that the denominator may become zero due to approximation errors in the trigonometric functions. Therefore, $r_{atan2}(y, x)$ is not suitable for RCF. However, by analyzing the behavior of $r_{atan2}(y, x)$, a new RCF can be found. When there is no approximation error in the trigonometric function, the problem of the denominator being zero does not occur because the numerator also becomes zero at that time, and the effect of the correction term disappears. From this, the requirement for RCF is that the denominator of $r_{atan2}(y, x)$, $f'(\theta)$ is zero when it is also zero. Therefore, $f'(\theta)$ itself is a candidate for RCF here.

$$
\tilde{r}_{atan2}(y, x) = \alpha_{atan2}[xs_2(\theta) - yc_2(\theta)]
\tag{14}
$$

Although there are design degrees of freedom in the coefficients, the result of this optimization is approximately 1.0 as shown next, thus eliminating one multiplication.

$$
\alpha_{atan2} = \arg\min_{\alpha} \int_{-\pi}^{\pi} |\text{at2}_*(y, x) - \tilde{r}_{atan2}(y, x) - \theta| d\theta \approx 1.0
\tag{15}
$$

The same result can be obtained using either $\text{at2}_{if}(y, x)$ or $\text{at2}_{ex}(y, x)$ as $\text{at2}_*(y, x)$. A plot of $\tilde{r}_{atan2}(y, x)$ is shown in Figure 6a. Here, we use the domain expansion $at2_{ex}(y, x)$ based on the half-angle identity. It can be seen that $\tilde{r}_{atan2}(y, x)$, which can be computed by the multiply–add operation, agrees very well with the residual of $\text{at2}_{ex}(y, x)$ and is appropriate as RCF.

Figure 6b shows the case where $\text{at2}_{if}(y, x)$ is evaluated with the same RCF, and this RCF can always be applied if a good approximation is obtained in the first approximation. Therefore, other atan2 approximation methods, such as DSP-Trick, for example, are also RCFs that can be made highly accurate.

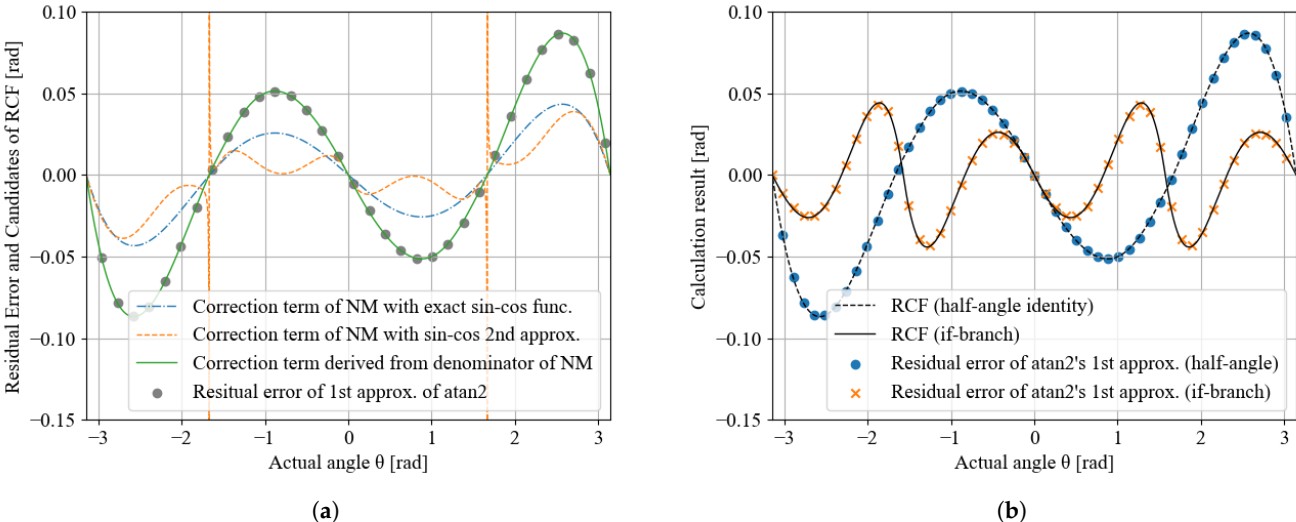

(**a**)                                                                                    (**b**)

**Figure 6.** Residual error of atan2's 1st approximation and consideration of its RCF candidates (plot using $x = \cos\theta, y = \sin\theta$). (**a**) Evaluation of Newton's method (NM) for RCF. (**b**) Applying desined RCF for each expansion.

### 4.3. Second Approximation of Inverse Trigonometric Functions

Having obtained the RCF, the second approximation of atan2 can be computed using the second approximation of trigonometric functions as follows:

$$\theta_1 = \text{at2}_*(y, x) \tag{16}$$

$$\text{at2}_2(y, x) = \theta_1 - \tilde{r}_{atan2}(y, x)$$
$$= \theta_1 - x\text{s}_2(\theta_1) + y\text{c}_2(\theta_1). \tag{17}$$

Figure 7 shows the residuals of $\text{at2}_{if}(y, x)$ and $\text{at2}_{ex}(y, x)$. Both $\text{at2}_{if}(y, x)$ and $\text{at2}_{ex}(y, x)$ can be used for $\text{at2}_*(y, x)$ in terms of error, but $\text{at2}_{if}(y, x)$ should be used in terms of the computational cost.

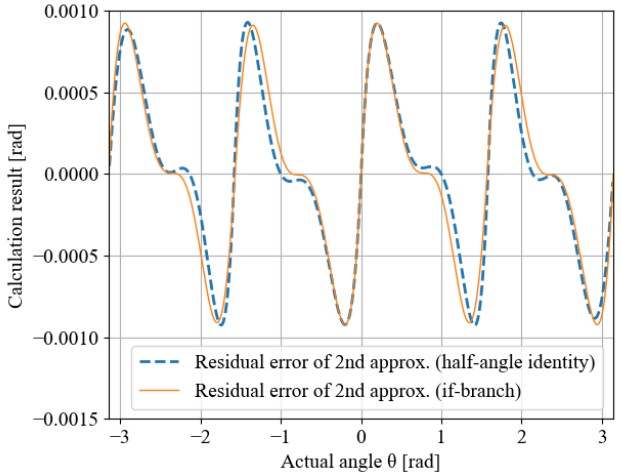

**Figure 7.** Residual error of 2nd approximation of atan2 (plot using $x = \cos\theta, y = \sin\theta$).

In addition, the constraint $x^2 + y^2 = 1$ was used in the approximate formula design up to this point, but normalization may be necessary in some applications. Our proposed atan2 approximation can be combined with any normalization algorithm. Recent CPUs can compute faster with the SIMD instruction set. However, for MCUs with only a simple instruction set, normalization is a major problem [49]. Therefore, fast inverse square root, which can be executed using only multiply–add and logic operations, is a useful technique

for MCUs. Any normalization algorithm or SSE instruction can be used in general; however, we assume an MCU and verify the use of fast inverse square root. The normalized second approximation is defined as $at2_{2n}(y, x)$ when the fast inverse square root is used for the variables. $at2_{2n}(y, x)$ can be calculated using the fast inverse square root $G(\cdot)$ as follows:

$$at2_{2n}(y, x) = at2_2(ny, nx), \tag{18}$$

$$\because n = G(x^2 + y^2) \approx \frac{1}{\sqrt{x^2 + y^2}} \tag{19}$$

The error between the calculation results and the Math library is shown in Figure 8. The increase in computational complexity due to normalization is shown as $at2_{2n}$ in Table 2. Compared with $at2_2$, it is two times greater, which indicates that a highly accurate approximation of $at2_2$ can be performed with the same computational complexity as that of the fast inverse square root.

**Table 2.** Calculation time evaluation of 2nd approximations of atan2 (Average$\pm$ S.D.)

| | 1st Approx. | 2nd Approx. $at2_{if}$/$at2_{ex}$ | 2nd Approx. with FISR $at2_{nif}$/$at2_{nex}$ | Math Library (math.h) |
|---|---|---|---|---|
| Calculation time | $1.43 \pm 0.26$ ns | $2.96 \pm 0.27$ ns<br>$4.76 \pm 0.26$ ns | $7.3 \pm 0.19$ ns<br>$9.14 \pm 0.25$ ns | $21.96 \pm 0.83$ ns |

In this study, the original fast inverse square root [32] is used for the evaluation implementation, but faster implementations have been studied, such as the algorithm [50], which is faster for FPGA.

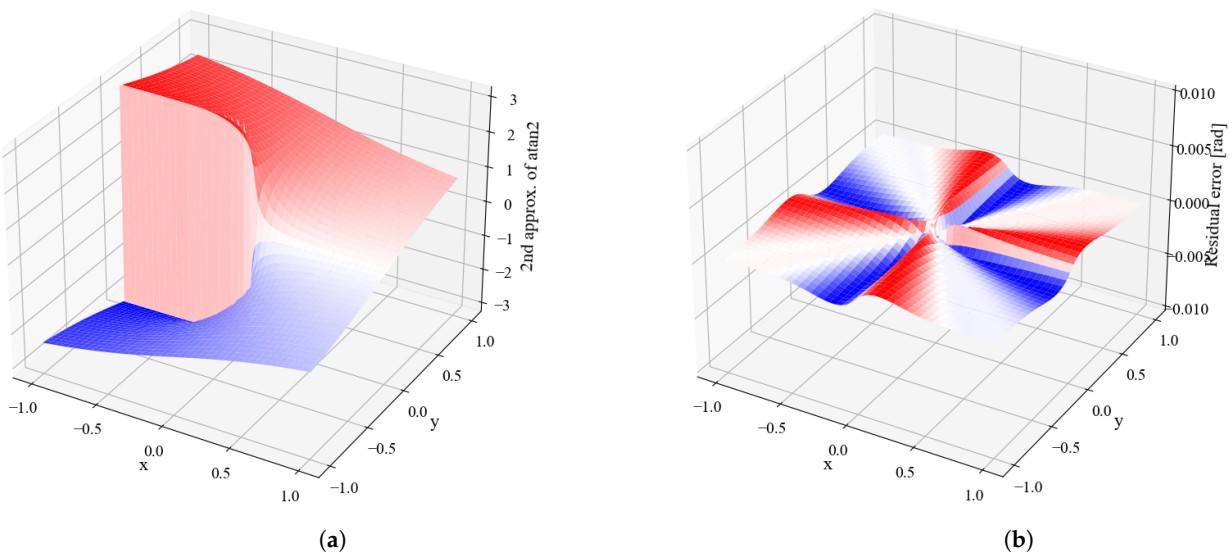

(**a**)　　　　　　　　　　　　　　　　　　　　　　(**b**)

**Figure 8.** Atan2's 2nd approximation with normalization using fast inverse square root(FISR) and its residual error. (**a**) Atan2 approx. over entire domain using FISR. (**b**) Error with atan2 in math library.

## 5. Results

Second approximations of the trigonometric and inverse trigonometric functions have been developed, and the results are summarized here. The maximum error indicates the maximum value of the final residuals of the second approximation. The computational cost indicates the number of multiply–add operations used. Although an example of the computation time has already been shown, only the number of operations is shown here because it varies depending on the CPU performance and architecture.

For the second approximation of the trigonometric functions, Equations (3) and (4) are considered good approximations, and the error and computational cost are as shown

in Table 3. The final error is less than 0.001 for the entire definition region, and the computational cost is very low.

**Table 3.** Error and computational cost of approximating trigonometric functions.

| Sine Function | Maximum Error | Computational Cost |
|---|---|---|
| RCM 1st approximation | $5.6 \times 10^{-2}$ | 1 addition<br>2 multiplications<br>1 absolute value |
| RCM 2nd approximation | $9.2 \times 10^{-4}$ | 2 additions<br>4 multiplications<br>2 absolute values |

Equation (17) is considered a good second approximation of the inverse trigonometric function. The error and computational cost are shown in Table 4. The small approximation error is less than 0.001 rad over the entire definition region. The computational cost is a little higher because two second approximations of trigonometric functions are required, but the approximation formula is only a finite number of multiply–add operations.

**Table 4.** Error and computational cost of inverse trigonometric functions.

| Atan2 Function | Maximum Error | Computational Cost |
|---|---|---|
| RCM 1st approximation | $4.2 \times 10^{-2}$ rad | 1 addition<br>2 multiplications |
| RCM 2nd approximation | $9.2 \times 10^{-4}$ rad | 7 additions<br>14 multiplications<br>4 absolute values |

## 6. Discussion

In this study, we have designed a second approximation of RCF to achieve a higher accuracy in approximating trigonometric and inverse trigonometric functions.

### 6.1. Approximation of Trigonometric Function

As mentioned in Introduction, the Taylor series is slow to converge but can compute mathematically exact values of trigonometric functions. We compare the Taylor series and the proposed method in the domain of definition $[-\pi, \pi]$. Taylor series are evaluated from the first to the ninth order. In addition, the computational cost is evaluated based on the definition formula and the following formula expansion to reduce the number of multiplications for embedding.

$$\sin(x) \approx \sum_{n=0}^{k} (-1)^n \frac{x^{(2n-1)}}{(2n-1)!} = x \left( 1 - x^2 \left( \frac{1}{3!} + x^2 \left( \frac{1}{5!} - x^2 \left( \frac{1}{7!} + \cdots \right) \right) \right) \right) \quad (20)$$

The left-hand side of Equation (20) is the definition for the kth-order Taylor expansion and the right-hand side is the optimized number of multiplications by storing $x^2$ in memory. The error is evaluated numerically by Romberg integration, and the computational cost is compared to the number of multiplications.

Table 5 and Figure 9 show the comparison results. As mentioned above, the Taylor series is mathematically exact, but its convergence is slow, so a large error remains if a wide domain of definition is used. Although the proposed method cannot reduce the error to zero, it enables obtaining results with a reduced number of calculations and reduced error.

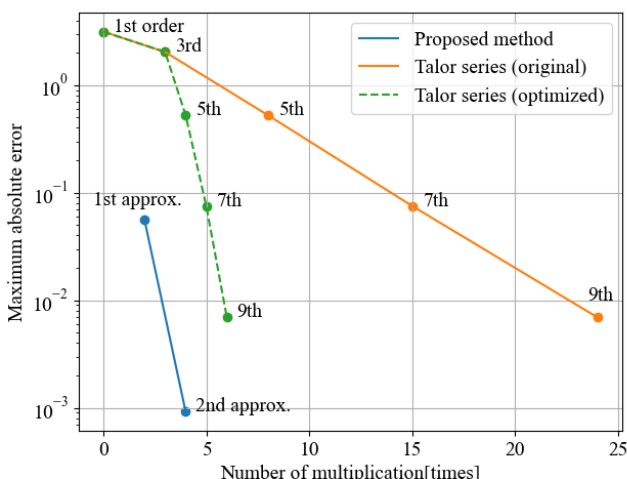

**Figure 9.** Comparison of maximum error and number of multiplications of sin function approximation between Taylor series and proposed method (log scale is used in vertical axis to compare error order).

**Table 5.** Comparison summary of sin function approximation between Taylor series and proposed method.

| | Taylor Series (Original/Optimized for Embedding) | | | | | Proposed Method | |
|---|---|---|---|---|---|---|---|
| | **1st Order** | **3rd Order** | **5th Order** | **7th Order** | **9th Order** | **1st Approx** | **2nd Approx** |
| Maximum abolute error | 3.14 | 2.03 | $5.24 \times 10^{-1}$ | $7.52 \times 10^{-2}$ | $6.93 \times 10^{-3}$ | $5.6 \times 10^{-2}$ | $9.2 \times 10^{-4}$ |
| Mean absolute error | 5.84 | 2.25 | $4.23 \times 10^{-1}$ | $4.80 \times 10^{-2}$ | $3.66 \times 10^{-3}$ | $1.9 \times 10^{-1}$ | $3.3 \times 10^{-3}$ |
| Number of multiplication | 0/0 | 3/3 | 8/4 | 15/5 | 24/6 | 2 | 4 |

### 6.2. Approximation of Inverse Trigonometric Function

Next, we discuss the approximation of the inverse trigonometric function. Depending on the characteristics of the application, the desired accuracy can be used by selecting whether the first approximation is sufficient, second approximation is necessary, or whether normalization is necessary. Table 6 shows a comparison with other methods. Basically, all are trade-offs between accuracy and computational complexity, and should be used appropriately depending on the application. The approximate formula by the residual correction method is an algebraic formula and is extremely easy to implement as a program, so it can be one of the options for approximate calculation. While more exact calculation methods are suitable for obtaining exact values, such as in numerical simulations, multiply–add operations are very effective in compact systems such as MCUs, FPGAs, and large-scale machine learning operations. In such fields, the proposed method should be used to obtain highly accurate approximations with a finite number of multiply–add operations without using division.

**Table 6.** Comparison with other methods.

| Atan2 Approx. Methods | Accuracy | Division | Iteration | Memory |
|---|---|---|---|---|
| Rational approx. [34] | High | Necessary | - | - |
| Tayler expansion | Depends on order | - | - | - |
| LUT | Depends on memory | - | - | Large |
| CORDIC [38] | Depends on iteration | Necessary | Necessary | - |
| DSP-trick [42] | Middle (<0.01) | Necessary | - | - |
| RCM 1st approx. | Middle (<0.01) | - | - | - |
| RCM 2nd approx. * | High (<0.001) | - | - | - |

*: The proposed methods in this study.

The following is an example of compilation and execution results on an actual MCU. The program memory occupancy of the proposed algorithm with an MCU grade is shown in Table 7. The famous Microchip PIC family and the XC8 compiler are considered in these examples. In the lowest grade model, the atan2 function alone consumes the entire program memory, and thus the program cannot be compiled. In contrast, the proposed method makes it possible to successfully implement the algorithm. Other grades of MCUs without FPU or DSP occupy the same amount of program memory. Since embedded systems usually require not only function calculations but also various processes, such as measurement and equipment control, the program memory of the MCU must be saved as much as possible in order to implement these processes. The second approximation of the proposed method also uses s2 and c2 in the calculation process; thus, trigonometric functions can be simultaneously used. A total of 7.7 KB is required to include them in the Math library, so the approximation is advantageous for performing complicated calculations.

Next, the calculation speed is measured using an actual MCU (PIC18F2580). The expetimental results are summarized in Table 8. The calculation speed is fast enough for the first approximation of atan2, and the speed of the second approximation is close to that of the Math library as the calculation volume increases. Therefore, if there is sufficient program memory, using the Math library is the most accurate way to calculate accurate values. However, for limited program memory, such as in low-grade MCUs, the first approximation can be used for speed and the second approximation can be used if accuracy is required. This increases the degree of freedom of implementation. As mentioned above, MCUs perform not only function calculations but also various other functions, such as measurement and equipment control, so function calculations should be as fast as possible.

**Table 7.** Comparison of occupied program memory compiled with actual MCUs.

| Grade | Example of MCU | Available Program Memory | Proposed Methods | | Math Library (math.h) | |
|---|---|---|---|---|---|---|
| | | | $at2_if$ 1 KB Required | $at2_2$ (w/s2 and c2) 1.8 KB req'd | atan2 Only 3 KB req'd | atan2, sin, cos 7.7 KB req'd |
| Low | PIC12F1572 | 2 KB | A | AR | CC | CC |
| Middle | PIC16F1827 | 4 KB | A | A | AR | CC |
| Middle | PIC16F1769 | 8 KB | A | A | A | AR |
| High | PIC18F2580 | 16 KB | A | A | A | A |

A: available (less than 50% occupancy); AR: available with restriction; CC: cannot be compiled.

**Table 8.** Experimental result on actual MCU (PIC18F2580@4MHz internal clock).

| | 1st Approx of atan2 | 2nd Approx of atan2 | atan2 (math.h) |
|---|---|---|---|
| Average time of 100 iterations | 1.5 ms | 6.5 ms | 8.0 ms |

In this study, we developed highly accurate approximation formulas for sin and cos as trigonometric functions and atan2 as an inverse trigonometric function using the

residual correction method. The tan, asin, and acos functions can be calculated from the fast inverse square root and the approximation formula proposed here using the trigonometric identities, respectively. Therefore, it is known that log and exp functions can be computed from the IEEE754 structure as the fast inverse square root, implying that, besides the four arithmetic operations, logarithm and exponential, trigonometric, and inverse trigonometric functions can also be computed quickly. This enables the computation of highly accurate approximations of all elementary functions by a finite number of multiply–add operations and bit shifts, when considered for embedded systems.

## 7. Conclusions

In this paper, we proposed a second approximation of the residual correction method that can approximate trigonometric and inverse trigonometric functions with a high accuracy using a small number of calculations. Using our proposed algorithm, nonlinear calculations that were previously impossible to implement can now be realized even on small, low-power computers, such as MCUs. It cannot reduce the error to completely zero; however, it can compute accurate approximations with low-cost computation. This can contribute to machine downsizing and cost reduction, and is a candidate algorithm for developing MCU-based devices that require real-time performance, such as drones and IoT devices.

To approximate trigonometric and inverse trigonometric functions with a high accuracy using finite multiply–add operations, a second approximation using the residual correction method was proposed. Trigonometric functions can be computed with a maximum error of less than 0.001 by two additions and four multiplications, and atan2 can be computed with a maximum error of less than 0.001 rad by seven additions and fourteen multiplications. For the formulation of the second approximation, we designed and evaluated a formula using floating-point arithmetic. In future research, we aim to optimize the formula for embedded systems by converting it to fixed-point arithmetic. Since the residual correction method is formulated using only the multiply–add operation without division, it is well suited to the fixed-point arithmetic, and further acceleration can be expected when considering implementation in embedded systems.

**Author Contributions:** Conceptualization, T.K.; investigation, T.K.; methodology, T.K.; supervision, T.T.; validation, T.K. and T.T.; writing—original draft, T.K.; writing—review and editing, T.T. All authors have read and agreed to the published version of the manuscript.

**Funding:** This research received no external funding.

**Institutional Review Board Statement:** Not applicable.

**Informed Consent Statement:** Not applicable.

**Data Availability Statement:** Not applicable.

**Conflicts of Interest:** The authors declare no conflict of interest.

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
