# Peer review of "Fast and Accurate Approximation Methods for Trigonometric and Arctangent Calculations for Low-Performance Computers"

_electronics, doi:10.3390/electronics11152285_

Round 1

Reviewer 1 Report

This paper presented the “Fast and accurate approximation methods for trigonometric and arctangent calculations for low-performance computers”. This work is valued on a highly accurate approximation in a small number of computation cycles, which may be helpful when embedded in an FPGA or microcontroller or when performing many large-scale operations on a CPU. It devises algorithms to obtain the desired calculated values without accurate approximations of certain non-linear functions with only a small number of multiply-add operations. This study aims to design an algorithm for the approximate computation of trigonometric and inverse trigonometric functions at a fast and accurate computation. The results can be used to stabilize the posture control for robots and drones. The contributions can be applied to speed up the learning of deep neural networks. This paper is relevant and valuable to the readers of Electronics. To guide you, I have several suggestions which I believe would improve your manuscript.

  1. All abbreviations must be written in full in their first appearance before using them in subsequent occurrences. For example, please show the full name for the below abbreviations and explain them.

(1) On page 1, line 22 to 23, please show FPGAs and FPUs.

(2) On page 2, line 56, please show CORDIC.

(3) On page 2, line 64, please show DSP.

  1. The value of this paper is not significant. This means that the authors shall demonstrate what the difference between this study and the others is. The authors may read more papers in this field and do dome examples to explain the differences from the other research to increase research values.
  2. In Section 2, the authors try to propose their concept function. However, I suggest they still need to cite some relative works or references. The references are lacking.
  3. In conclusion, the conclusion is more like a project report but listed some suggestions without giving a benefits discussion. Are there any comments after finding something? How employ these findings? Do the authors provide any solutions to improve the FPU?  

Reviewer 2 Report

The authors proposed a novel way to approximate the calculations of trigonometric and arctangent for low-performance computers. The results look promising, but I hope authors could provide some clarification regarding to my doubts.

1.     Authors mentioned this method is good for low- performance computers, but what are these typees of low- performance computers which are still used currently? Maybe it is better to write this in the introduction.

2.     Authors mentioned that Eqs. (1-2) and Eq.(5) are from previous study. Could authors provide the reference of this study?

3.     For the RCF function, the “rc” is signed in the left top of function. Is this a mathematically reasonable format?

4.     Are there other existing approximation approaches from the literatures? Could authors provide a table to compare the computational time between the approximation of functions using Tylor expansion with the proposed method, and existing methods from the literatures?

5.     While the performance of proposed method is good, does the reduced computational time really matter in these computers? Please explain.

Reviewer 3 Report

There are following some suggestions/comments

1.     There are many grammatical, typographical errors and inconsistent mathematical expressions in the paper. They can be easily seen from the text. Read carefully.

2.     What is your main contribution? The contribution of the current work should be emphasized in the introduction.

3.     What kind of reason sent you to study this topic?

4.     What are the limitations and benefits of your work?

5.     More discussions are needed to be detailed in numerical results/examples.

6.     The abstract, conclusion and introduction need to be re-written/revised properly, in terms of the suggestions.

7.     Also add the CPU time of algorithm.

8.     All figures are needed in more attractive way.

9.     Give some examples of applications of your own work not need to give general examples. Where your study is work and gives some benefits to your country economy.

Reviewer 4 Report

CONTENT

=======

The paper is about developing fast approximations for trogonometric functions (sin, cos, arctan).

Based on a previous study of the same authors, this article develops new approximations using only

add and multiply operations, and the performance of the proposed algorithms are presented

(in terms of numerical error).

EVALUATION

==========

While the results are clearly presented and some empirical evaluations have been carried out,

I can see the following issues:

- The background literature survey could be more extensive and more detailed. For example, the fast inverse square root algorithm (from e.g., Quake 3) plays a crucial role in the algorithms - has it been implemented and used already in

other software / hardware? If so, how was the performance? Similarly, existing methods for approximating trigonometry functions are only mentioned very briefly without real comparisons with the proposed approach.

- The focus of the paper is about avoiding the use of "difficult" operations such as division. How does this affect

the actual performance of the algorithms? In the experiments section we can only see some plots about the errors, but not the actual performance comparisons with existing algorithms.

- Some formulas are used without much explanation or derivation, e.g., (1) and (5) - I suppose these must be well-known, but please cite some sources or at least briefly mention how they are derived.

- Some text are speculative with little analysis to support them, e.g., l. 120, "computation time is approximately twice that for the second approximation, which is reasonable 121

since the parabolic function is computed twice for the second approximation.".

- Consistency - some figures use linear scales while some others use log scales.

Round 2

Reviewer 1 Report

This paper presented the “Fast and accurate approximation methods for trigonometric and arctangent calculations for low-performance computers”. This work is valued on a highly accurate approximation in a small number of computation cycles, which may be helpful when embedded in an FPGA or microcontroller or when performing many large-scale operations on a CPU. It devises algorithms to obtain the desired calculated values without accurate approximations of certain non-linear functions with only a small number of multiply-add operations. This study aims to design an algorithm for the approximate computation of trigonometric and inverse trigonometric functions at a fast and accurate computation. The results can be used to stabilize the posture control for robots and drones. The contributions can be applied to speed up the learning of deep neural networks.

I appreciate the authors' efforts on revising the manuscript. All issues I concerned that have been addressed by the authors. Thank you.

Reviewer 2 Report

The revision has improved the quality of this paper.